# Numerical evaluation reveals the effect of branching morphology on vessel transport properties during angiogenesis

Fatemeh Mirzapour-Shafiyi[1,2☯], Yukinori Kametani[3☯], Takao Hikita[1], Yosuke Hasegawa[3☯]*, Masanori Nakayama[1,2,4☯]*

**1** Max Planck Institute for Heart and Lung Research, Laboratory for Cell Polarity and Organogenesis, Bad Nauheim, Germany, **2** DFG Research Training Group, Membrane Plasticity in Tissue Development and Remodeling, Philipps-Universität Marburg, Marburg, Germany, **3** Institute of Industrial Science, The University of Tokyo, Tokyo, Japan, **4** Kumamoto University International Research Center for Medical Science, Kumamoto, Japan

☯ These authors contributed equally to this work.
* ysk@iis.u-tokyo.ac.jp (YH); masanori.nakayama@mpi-bn.mpg.de (MN)

**Data Availability Statement:** All relevant data are within the manuscript and its Supporting Information files.

## Abstract

Blood flow governs transport of oxygen and nutrients into tissues. Hypoxic tissues secrete VEGFs to promote angiogenesis during development and in tissue homeostasis. In contrast, tumors enhance pathologic angiogenesis during growth and metastasis, suggesting suppression of tumor angiogenesis could limit tumor growth. In line with these observations, various factors have been identified to control vessel formation in the last decades. However, their impacts on the vascular transport properties of oxygen remain elusive. Here, we take a computational approach to examine the effects of vascular branching on blood flow in the growing vasculature. First of all, we reconstruct a 3D vascular model from the 2D confocal images of the growing vasculature at postnatal day 5 (P5) mouse retina, then simulate blood flow in the vasculatures, which are obtained from the gene targeting mouse models causing hypo- or hyper-branching vascular formation. Interestingly, hyper-branching morphology attenuates effective blood flow at the angiogenic front, likely promoting tissue hypoxia. In contrast, vascular hypo-branching enhances blood supply at the angiogenic front of the growing vasculature. Oxygen supply by newly formed blood vessels improves local hypoxia and decreases VEGF expression at the angiogenic front during angiogenesis. Consistent with the simulation results indicating improved blood flow in the hypo-branching vasculature, VEGF expression around the angiogenic front is reduced in those mouse retinas. Conversely, VEGF expression is enhanced in the angiogenic front of hyper-branching vasculature. Our results indicate the importance of detailed flow analysis in evaluating the vascular transport properties of branching morphology of the blood vessels.

## Author summary

Blood vessels are important for the transport of various substances, such as oxygen, nutrients, and cells, to the entire body. Control of blood vessel formation is thought to be

**Funding:** Funding for this project was provided by the German Research Foundation, Deutsche Forschungsgemeinschaft (GRK2213) and Excellence Cluster Cardio-Pulmonary system (https://www.cpi-online.de) for MN, TH and FMS. The JSPS KAKENHI Grant Number JP17H03170 and JP17KK0128 for YK and YH. The funders had no role in study design, data collection and analysis, decision to publish, or preparation of the manuscript.

**Competing interests:** The authors have declared that no competing interests exist.

important in health and disease. In the last decades, various factors which regulate blood vessel branching morphology have been identified. Gene modification of some of these identified factors results in hyper-branching of the vasculature while others cause hypo-branching of the vessel. Given the importance of the transport property of the blood vessel, it is important to examine the effect of these identified factors on the transport property of the affected vascular morphology. In line with these facts, we reconstruct 3D vessel structures from 2D confocal microscopy images. We then numerically simulate blood flow in the structures. Interestingly, our results suggest vessel network complexity negatively affects the blood perfusion efficiency and tissue oxygenation during angiogenesis. Thus, our results highlight the importance of flow analysis considering the detailed 3D branching pattern of the vascular network to quantitatively evaluate its transport properties.

## 1. Introduction

Upon tissue hypoxia, proangiogenic factors such as vascular endothelial growth factors (VEGFs) are secreted to induce new blood vessel formation from existing vessels, termed angiogenesis [1]. While it is important to promote angiogenic vessel growth for tissue homeostasis, the vessel formation in tumors can give transformed cells better access to nutrients and oxygen [2–4]. Transformed cells hijack blood vessels for metastasis to a distant tissue [5]. In the last decades, various studies have identified critical factors controlling endothelial proliferation and vascular branching [6–10]. In line with these observations, anti-angiogenic therapy aimed to starve tumor cells of nutrients and oxygen by reducing tumor vascularization. However, the outcome of the treatment was more limited than expected [11,12]. Reducing blood vessel formation in tumor tissues is thought to enhance ischemia to induce tumor resistance against chemotherapies as well as to restrict drug delivery [6,13].

Recently, numerical simulation of blood flow and associated transport phenomena in biological systems attracts attentions, since it has a potential to clarify the complex dynamics of blood flow within complex capillary network, and also provide quantitative information on local hemodynamic parameters such as wall shear stress, pressure and oxygen concentration, which are difficult to obtain experimentally *in vivo*. However, attempts to compare transport properties of vascular networks with different morphological features are still quite limited. This could be achieved by artificially removing/adding vessels from a reference structure based on statistical or empirical rules [14,15]. However, the resultant structures are no longer real, and experimental validation is prohibitive.

In the present study, we apply genetic modification techniques to realize hyper- and hypo-branching vascular network in mouse retina. This allows us to systematically analyze blood flow and associated oxygen transport for the real structures with different morphological features. The obtained numerical results are also compared with experimental observation for validation. Interestingly, hyper-branching morphology attenuates effective blood flow at the angiogenic front *in silico* and fails to improve tissue hypoxia *in vivo*. In contrast, hypo-branching morphology enhances blood supply at the growing vasculature. Consistently, VEGF expression of the angiogenic front region is efficiently improved, suggesting better oxygen supply at the region. Our results indicate the importance of evaluating branching networks of the vasculature by transport property of the blood vessel.

## 2. Materials and methods

### 2.1. Ethics statement

All animal experiments were conducted according to the protocols approved by the local animal ethics committees and authorities (Regierungspräsidium Darmstadt, B2/1073) and institutional regulations.

**2.1.1. Mice breeding.** As previously described, transgenic *Pdgfb*-iCre mice were bred into lines of animals containing a *LoxP*-flanked *Prkci* [16] and *LoxP*-flanked *Foxo1* [17]. Intraperitoneal injections of tamoxifen (Sigma, MO, T5648) from postnatal day1 (P1) to P3 were used to induce activation of Cre in neonatal mice. The phenotype of the mutant mice was analyzed at P5 and tamoxifen injected Cre negative littermates were used as controls.

### 2.2. Retina staining

For retina staining, eyeballs were fixed for 20 min in 2% Paraformaldehyde (PFA; Sigma, P6148) at room temperature (RT). Afterwards, retinas were dissected in PBS and fixed for 30min in 4% PFA on ice. Next, they were washed three times with PBS and permeabilized and blocked for 2hr at RT in blocking buffer (BB): 1% fetal bovine serum (FBS; Biochrom GmbH, Berlin, Germany), 3% Bovine Serum Albumin (BSA; Sigma, A2153), 0.5% TritonX-100 (Sigma, T8787), 0.01% Na deoxycholate (Sigma, D6750) and 0.02% Na Azide (Sigma, S8032) in PBS on rocking platform. Then they were incubated with primary antibodies (anti-ICAM-II (BD Pharmingen, 553326, 1:100), anti- Collagen-Type IV (Collagen-IV) (Bio-RAD, 2150–1470, 1:400) and anti-VEGF164 (R&D Systems, AF-493-NA, 1:100) in 1:1 BB/PBS), overnight at 4°C on rocking platform. Retinas were then washed four times for 30 min in PBS/ 0.2% TritonX-100 (PBT) at RT and incubated with Alexa Fluor conjugated secondary antibodies (Invitrogen, 1:500) in 1:1 BB/PBS for 2hr at RT. After another four times of washing with PBT, retinas were radially cut into four lobes and flat-mounted onto slides using Fluoromount-G mounting medium (Southern Biotech, 0100–01).

### 2.3. Image acquisition, processing and statistical analysis of vascular network

As the models for the hypo- and hyper-branched vasculature, we employed *lox*P-flanked *Prkci* [16] and *lox*P-flanked *Foxo1* [17] mice crossed with *pdgfb-icre* mouse. Endothelial proliferation is modulated via controlling c-Myc expression in both model mice [17,18]. After tamoxifen injection from P1 to P3 to induced effective gene deletion, mouse retina was harvested at P5 and stained with an anti-ICAM-II antibody the marker of the inner lumen of the blood vessels. Stained vasculatures were visualized by confocal microscopy. In the present study, a subset of the retinal tissue, consisting of a single artery as the flow inlet, a single vein as the flow outlet, and the whole in-between capillary bed, from the optic nerve head (ONH) at the center of retinal tissue to the angiogenic front of the vasculature, was extracted for further detailed analyses as shown in S1 Fig. This specific wedge-shaped region between an artery and a vein, which is referred to as 'A-V region' hereafter, can be considered as a minimum flow unit of the entire retina vasculature. The validity and limitation of analyzing the minimum flow unit are discussed in S1 Text.

To measure the property of the vasculature, morphometric measurements were introduced (Fig 1A). Consistent with the previous report [18], the retinal vasculature in the *Foxo1* EC specific inducible knock out (*Foxo1*[iΔEC]) showed increased vascular density, vessel length density and branching index (Fig 1B, 1D, 1E and 1F), while the same factors in the *Prkci* EC specific inducible knock out (*Prkci*[iΔEC]) were decreased (Fig 1C, 1G, 1H and 1I).

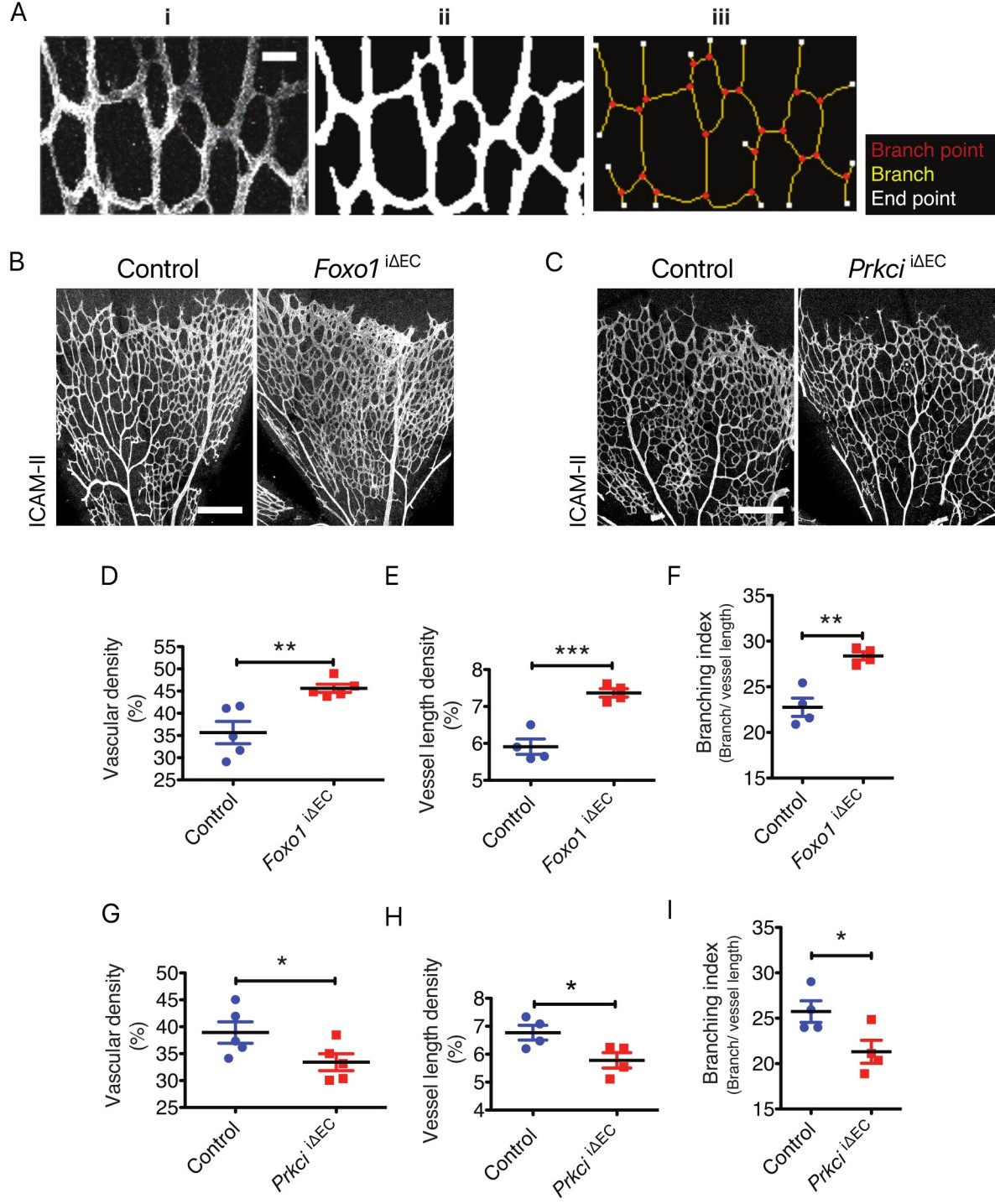

**Fig 1. Characterization of the hyper- and hypo-branching vascular models. A)** Grayscale image of a sample subset of the retinal microvasculature stained for ICAM-II (i), converted into the vessel mask (ii) and used for skeleton analysis (iii); Scale bar represents 30 μm. **B)** Staining of ICAM-II in control and $Foxo1^{i\Delta EC}$ mouse retinae at postnatal day 5 (P5); Scale bar represents 200 μm. **C)** Staining of ICAM-II in control and $Prkci^{i\Delta EC}$ mouse retinae at P5; Scale bar represents 200 μm. **D)** Quantification of vascular density (percentage of area occupied by vessels), **E)** vessel length density (percentage of area occupied by skeletonized vessel) and **F)** branching index (branch point/ mm of vessel length) in control and $Foxo1^{i\Delta EC}$ retina at P5. **G)** Quantification of vascular density, **H)** vessel length density and **I)** branching index in control and $Prkci^{i\Delta EC}$ mouse retinae at P5. Data represent mean ± S.E.M. two-tailed unpaired t-test $^*p < 0.05$, $^{**}p < 0.01$, $^{***}p < 0.001$ (n = 5 for D and G, n = 4 for E, F, H and I).

In the experiments using KO animal models, data are derived from three independent experiments (three sets of mutant mice and control littermates). The data are presented as mean ± S.E.M. All statistical analyses were carried out using Prism software (GraphPad, CA); $p < 0.05$ was considered as significantly different. Quantification of VEGF-A signal intensity was carried out by measuring mean gray value using ImageJ (1.52o). Morphometric analyses of the retinal vasculature were assessed using Fiji (2.0.0-rc-69/ 1.52n) Vessel Analysis plugin, Fiji Skeletonize plugin and Fiji Skeleton Analyzer. Volocity (Perkin Elmer, MA), Photoshop CS, Illustrator CS (Adobe), ImageJ and Fiji software were used for image processing in compliance with general guidelines for image processing.

## 2.4. 3D reconstruction of vascular network

The three-dimensional (3D) vascular models were reconstructed from the two-dimensional (2D) confocal images. The obtained RGB images were converted to black and white binarized images ('vessel mask') using MATLAB Image Processing Toolbox (Fig 2A), where white and black pixels correspond to the regions of the blood vessel and the surrounding tissue, respectively. The binarized vessel structure was then projected onto a $x$-$y|_{z = 0}$ plane in the 2D Cartesian coordinate system (Fig 2B). For all grid points within the blood vessel, the shortest distance to the vessel wall was calculated in the 2D structure, and then a 3D sphere with the diameter obtained at each grid point was placed in a 3D Cartesian coordinate system as schematically shown in Fig 2C. The envelope of all the spheres was then used to define the 3D blood vessel structure (Fig 2D). After reconstructing the 3D vascular structure, a signed distance function, which is commonly referred to as a level-set function [19], was computed at every grid point in both vessel and tissue regions. The obtained level-set function was integrated to an in-house solver for the blood flow.

## 2.5. Numerical simulation of blood flow

As a suspension of erythrocytes in plasma, blood exhibits shear-thinning properties due to rouleaux formation [20]. Rouleaux, aggregation of red blood cells (RBC), causes increased blood viscosity due to increased effective volume of RBCs [20]. Thus, blood behaves as a non-Newtonian fluid as its viscosity decreases with applied shear. However, under a sufficiently high shear rate ($> 150$ s$^{-1}$, [21]), blood flow can be accurately modeled as Newtonian fluid with a constant viscosity [22]. Nagaoka et al. measured averaged values of 632 to 1539 s$^{-1}$ for shear rate in human retinal first arteriole and venule branches [23]. Moreover, Windberger et al. [24] measured species-specific effect of shear rate on blood viscosity and erythrocytes aggregation within low shear rates of 0.7, 2.4 and 94 s$^{-1}$. They reported different viscosity values and RBC aggregation index among different mammalian species: horse, pig, dog, cat, rat, cattle, sheep, rabbit and mouse. In low shear rate regime (0.7 and 2.4 s$^{-1}$), they found lower shear dependent viscosity enhancement in cattle, sheep, rabbit and mouse, as compared to horse, rat, pig, dog and cat. Compared to other species, erythrocyte aggregation (EA, measured using four different methods) in mouse was found low and, in some methods, undetectable. At high shear rate (94 s$^{-1}$) they found the EA destroyed and the RBCs orientated to the flow direction. Based on these results, a simple Newtonian behavior is accounted to retinal flow in this model.

*In vivo* measurements have shown systolic and diastolic flow rate variations in human [25] and mouse [26] retinal vasculature. The influence of pulsation on flow is governed by the Womersley number [27]. This dimensionless number is defined as the ratio between the time-scale for the wall information propagating to the bulk fluid via fluid viscosity and the pulsation period. Consequently, when the Womersley number ($\alpha$) is sufficiently low ($\leq 1$), there is

                    

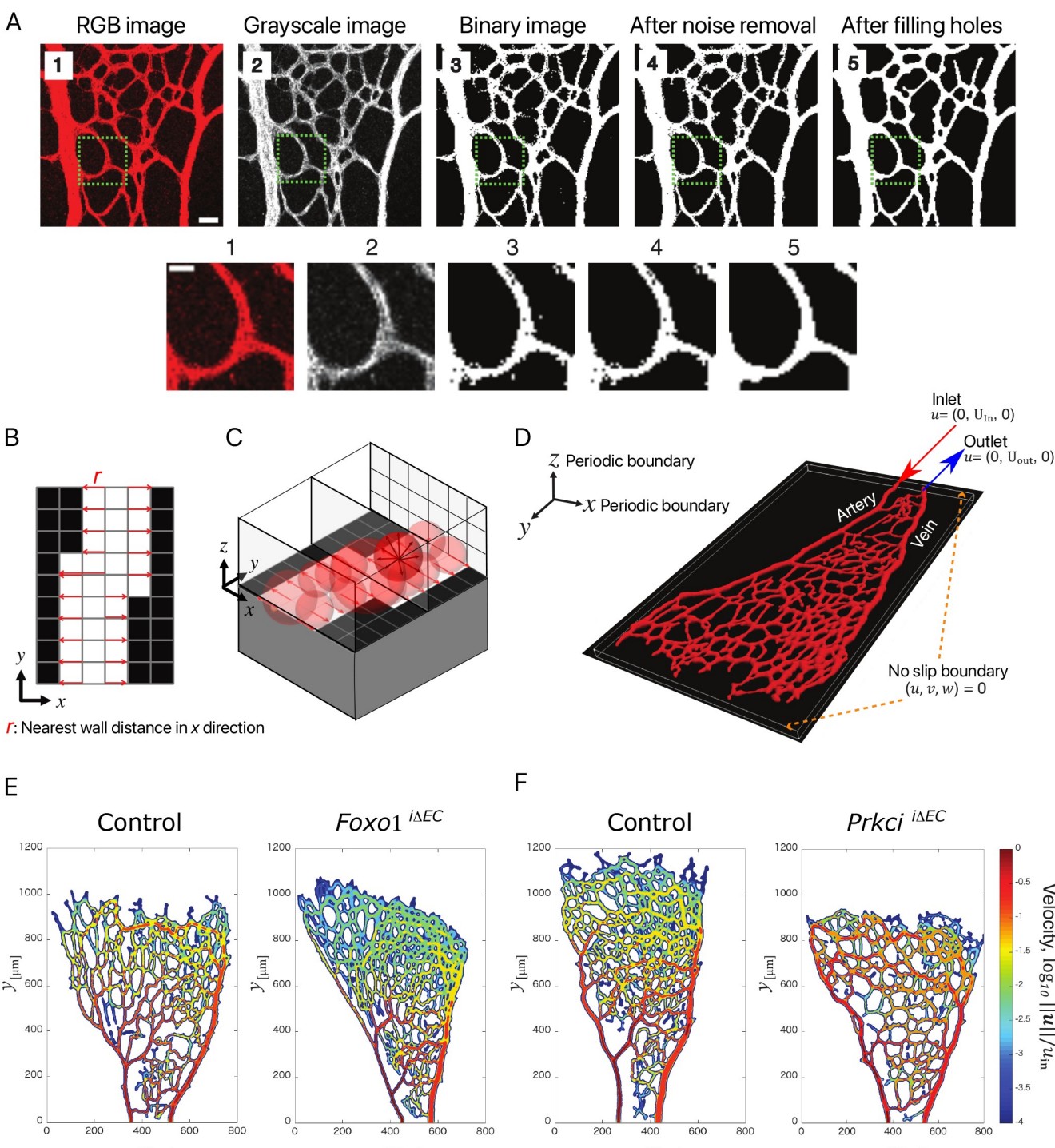

**Fig 2. Image processing and blood flow simulation from the growing mouse retinal vasculature. A)** Generation of binary vessel mask from RGB images. Higher magnification images of indicated areas in the upper panels are shown in the lower panels. Scale bar in the upper panel, 30 μm; 10 μm in the lower panel. **B)** Schematic of the searching process for the nearest solid wall (black pixel) on each pixel point and defining the local radius $r$ (pixel distribution is projected on Cartesian grid coordinate $\hat{x} = [x, y]$). **C)** The local radius $r$ for creating 3D vascular structure in a 3D Cartesian coordinate system, $x = [x, y, z]$, extended from $\hat{x}$. The region with $\|x\| < r$ is defined as blood vessel, while the rest is a surrounding tissue. **D)** Sample reconstructed 3D model of the retinal vasculature with indication of the inlet and outlet in Cartesian coordinate system. **E)** Visualization of the velocity amplitude on the central $x$-$y$ plane (along the $z$ axis) for $Foxo1^{i\Delta EC}$ and **F)** $Prkci^{i\Delta EC}$ mutants and respective controls; Color scale represents the logarithmic form of the normalized velocity ($\log_{10}\|u\|/u_{in}$).

enough time for a velocity profile to develop during each cycle (viscous-dominated flow), so that the resultant flow can be considered as a quasi-steady flow. While, for large values of $\alpha$ ($\geq 10$), the transient interaction between pulsation and blood flow plays an essential role, and thereby the unsteady flow analysis is crucial. The Womersley number is defined by the following formula:

$$\alpha = d\left(\frac{\omega}{v}\right)^{\frac{1}{2}} \tag{1}$$

Where $d$, $\omega$ and $v$ are the vessel diameter, the angular frequency for a heart rate and the kinematic fluid viscosity, respectively. Using *in vivo* reported values for mouse as such: $d \sim 10^{-5}$ [m], $\omega \sim 1$ [s$^{-1}$], and $v \sim 10^{-6}$ [m$^2$/s] [28–30], in neonatal mouse retina, $\alpha$ becomes as low as $\sim 10^{-2}$. Hence, the flow in retina model can be considered in viscous-dominated regime and this validates the quasi-steady assumption mentioned above. The other blood rheological properties such as the Fåhræus–Lindqvist effects, and the elastic effects of a vessel wall are not taken into account in the present model. Consequently, the flow is assumed to be incompressible, Newtonian and steady. The governing equations for the blood flow are given by the following Navier-Stokes and continuity equations:

$$u_j \frac{\partial u_i}{\partial x_j} = \frac{1}{Re} \frac{\partial^2 u_i}{\partial x_j \partial x_j} - \frac{\partial p}{\partial x_i} - \eta u_i \phi, \tag{2}$$

$$\frac{\partial u_i}{\partial x_i} = 0. \tag{3}$$

Here, $u_i$, $p$ and $Re$ denote the velocity component in the $i$-th direction, the static pressure and the Reynolds number, respectively. The inlet bulk mean velocity ($U_{in}^*$) and the diameter of the inlet artery ($D_{in}^*$) were used for non-dimensionalization in Eqs (2 and 3). Here, the superscript of $*$ represents a dimensional value, whereas a quantity without a superscript indicates a dimensionless quantity. The Reynolds number, expressing the ratio of the inertial and viscous forces, is defined as: $Re \equiv \frac{U_{in}^* D_{in}^*}{v^*}$. In this study, $Re$ was set to be 0.1 based on $D_{in}^*$ and the kinematic viscosity and a typical blood flow velocity in an artery reported in [24,28].

The current code is based on a volume penalization method [31], which is categorized as immersed boundary techniques [32]. The main advantage of the present approach is that an arbitrary 3D structure can be embedded in 3D Cartesian computational grids, so that there is no need to generate boundary-fitted grids for each geometry. Meanwhile, it has relatively slow convergence due to smearing of the fluid-solid boundary, so that we performed grid convergence studies to confirm that the present conclusions are not affected by further grid refinement (see, S2 Text and S2 Fig for the results of the grid convergence study). The last term on the right-hand-side of Eq (2) corresponds to an artificial body force term introduced in the volume penalization method (VPM) in order to realize a no-slip condition at a fluid-solid boundary [31]. Computational grid points were uniformly distributed in space and the spatial grid resolutions were set to be $(\Delta_x^*, \Delta_y^*, \Delta_z^*) \approx (1.5\ \mu m,\ 1.5\ \mu m,\ 1.5\ \mu m)$. The dimensions of the computational domain for each model are listed in Table 1. The dimensions of the computational domain depend on each sample, while the computational resolutions were kept constant for all the cases.

The computational domain is schematically shown in Fig 2D. Uniform velocity profiles were applied at the inlet and outlet boundaries, $U_{in}$ and $U_{out}$, respectively. The velocity at the outlet was determined such that the fluid volume is strictly conserved throughout the

**Table 1. Domain size and grid resolution for every model structure.**

| Name | Domain size [μm] $L_x^* \times L_y^* \times L_z^*$ | $D_{in}^*$ [μm] | Grid size [μm] | Number of grid points $N_x \times N_y \times N_z$ |
|---|---|---|---|---|
| *Foxo1* CTRL-1 | $800.30 \times 1072.10 \times 33.22$ | 12.08 | 1.51 | $530 \times 710 \times 22$ |
| *Foxo1* $^{iΔEC}$-1 | $755.00 \times 1102.30 \times 36.24$ | 12.08 | 1.51 | $500 \times 730 \times 24$ |
| *Foxo1* CTRL-2 | $588.90 \times 936.20 \times 36.24$ | 12.08 | 1.51 | $390 \times 620 \times 24$ |
| *Foxo1* $^{iΔEC}$-2 | $619.10 \times 694.60 \times 36.24$ | 10.57 | 1.51 | $410 \times 460 \times 24$ |
| *Foxo1* CTRL-3 | $694.60 \times 1087.20 \times 30.20$ | 13.59 | 1.51 | $460 \times 720 \times 20$ |
| *Foxo1* $^{iΔEC}$-3 | $936.20 \times 1011.70 \times 33.22$ | 13.59 | 1.51 | $620 \times 670 \times 22$ |
| *Prkci* CTRL-1 | $664.40 \times 1208.00 \times 36.24$ | 13.59 | 1.51 | $440 \times 800 \times 24$ |
| *Prkci* $^{iΔEC}$-1 | $830.50 \times 966.40 \times 33.22$ | 15.10 | 1.51 | $550 \times 640 \times 22$ |
| *Prkci* CTRL-2 | $845.60 \times 1253.30 \times 36.24$ | 13.59 | 1.51 | $560 \times 830 \times 24$ |
| *Prkci* $^{iΔEC}$-2 | $543.60 \times 1208.00 \times 30.20$ | 10.57 | 1.51 | $360 \times 800 \times 20$ |
| *Prkci* CTRL-3 | $588.90 \times 951.30 \times 33.22$ | 12.08 | 1.51 | $390 \times 630 \times 22$ |
| *Prkci* $^{iΔEC}$-3 | $588.90 \times 875.80 \times 27.18$ | 10.57 | 1.51 | $390 \times 580 \times 18$ |
| Cases with finer resolution for verification of grid resolutions | | | | |
| *Foxo1* CTRL-1F | $800.30 \times 1072.10 \times 33.22$ | 12.08 | 0.76 | $1060 \times 1420 \times 44$ |
| *Foxo1* $^{iΔEC}$-1F | $755.00 \times 1102.30 \times 36.24$ | 12.08 | 0.76 | $1000 \times 1460 \times 48$ |
| *Prkci* CTRL -1F | $664.40 \times 1208.00 \times 36.24$ | 13.59 | 0.76 | $880 \times 1600 \times 48$ |
| *Prkci* $^{iΔEC}$-1F | $830.50 \times 966.40 \times 33.22$ | 15.10 | 0.76 | $1100 \times 1280 \times 44$ |

structure. At the outer boundaries of the computational domain, a Neumann boundary condition was imposed for the pressure.

For spatial discretization, an energy conservative second-order central finite difference scheme (CDS) was applied for the convection term on the left-hand-side of Eq (2) on the staggered grid [33]. The diffusion term on the right side of Eq (2) was also discretized by a conventional second-order central finite difference scheme. For temporal advancement, the Simplified Marker and Cell, SMAC, [34] method was implemented to decouple the pressure in the Navier-Stokes equations shown in Eq (2). The second-order Crank-Nicolson scheme was employed for the diffusion and VPM terms. The other terms were calculated explicitly with the third-order Runge-Kutta scheme [35].

## 3. Simulation results

The obtained velocity fields showed distinct flow distributions for hyper- and hypo-branched structures (Fig 2E and 2F). In the present study, we conducted flow simulations for three independent samples for hyper- and hypo-branched structures, respectively. For reference, the results of the other two samples are also shown in S3 and S4 Figs. In general, the flow distribution was un-even in the *Foxo1* $^{iΔEC}$ vessel network comparing to that in control. Specifically, the velocity around the peripheral edge (angiogenic front) is drastically attenuated due to the hyper-branching. Conversely, the hypo-branched *Prkci* $^{iΔEC}$ network results in a more evenly distributed flow throughout the entire network including the peripheral edge.

As shown in Figs 2D and S1, the blood is supplied from an artery and distributed to the azimuthal direction by branching capillary network and then eventually flows into a draining vein toward the outlet. Hence, the spatial distribution of the azimuthal flow rate can be considered as a key quantity to evaluate the transport properties of the vascular network. In order to quantify the amount of flow transported to the peripheral regions, we introduced a cylindrical coordinate system ($r$ -$\theta$) with its origin at the first branching point from either the artery or the vein near the inlet (Fig 3A). Here, $r$ and $\theta$ represent the distance from the origin and the

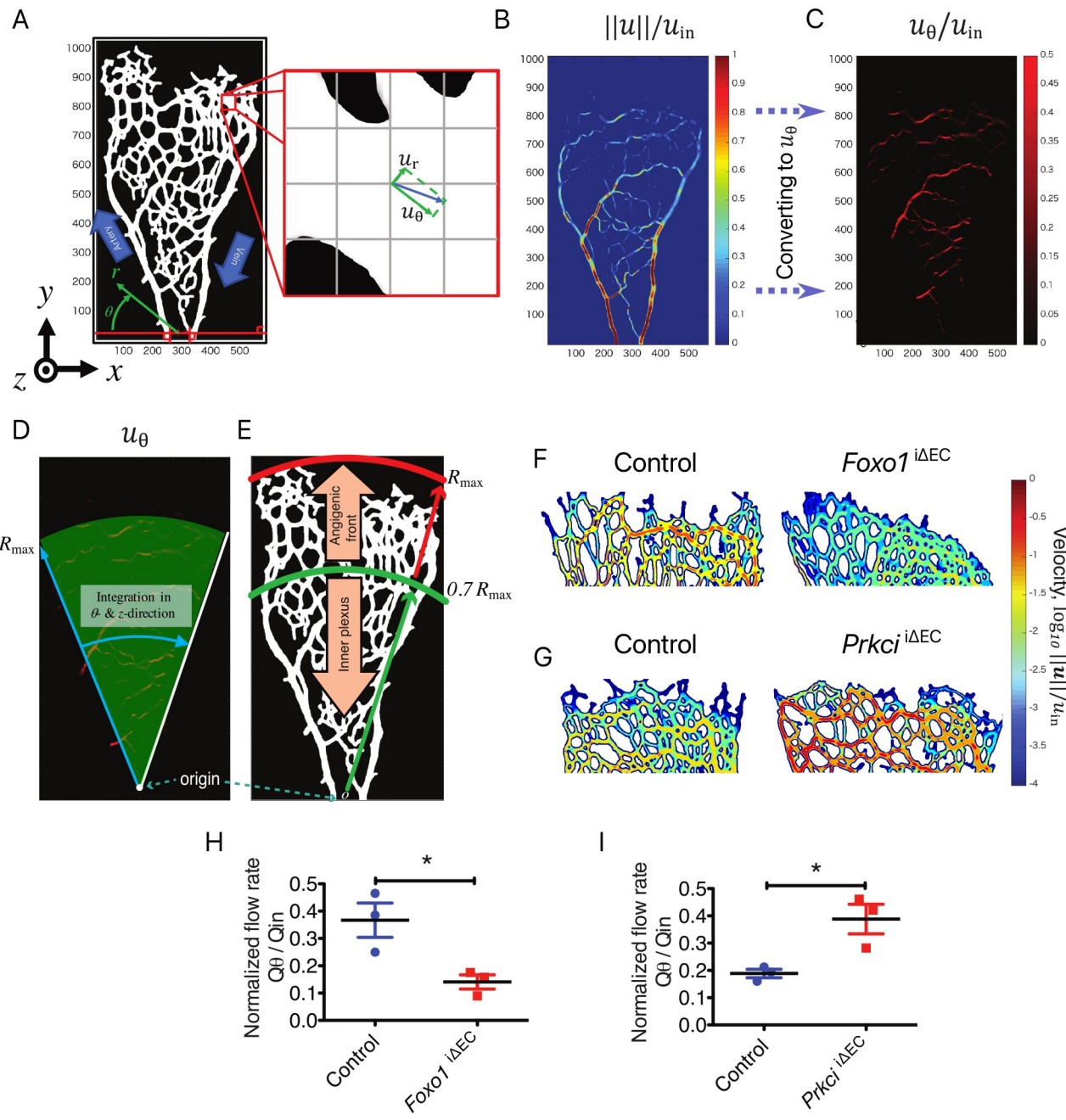

**Fig 3. Quantification of blood flow at the angiogenic front. A)** Coordinate transformation from the Cartesian system $(x, y)$ to the cylindrical system $(r, \theta)$. **B)** The velocity field transformed from the Cartesian coordinate system to the **C)** cylindrical one. **D)** Distribution of the azimuthal velocity $u_\theta$ between the artery and the vein. **E)** Schematic of defining the angiogenic front. The entire vasculature is separated into two regions: the inner plexus and the angiogenic front. **F)** Visualization of the velocity amplitude on the central $x$-$y$ plane (along the $z$ axis) in $Foxo1^{i\Delta EC}$ and **G)** $Prkci^{i\Delta EC}$ mutant mice and respective controls at P5; Color scale represents the logarithmic form of the normalized velocity ($\log_{10}\|u\|/u_{in}$). **H, I)** The normalized azimuthal flow rate at the angiogenic front of the $Foxo1^{i\Delta EC}$ (H) and $Prkci^{i\Delta EC}$ mice (I) with control. Data represent mean ± S.E.M. two-tailed unpaired t-test *p < 0.05 (n = 3).

azimuthal direction, respectively. Accordingly, the local flow velocity vector was decomposed into the radial and azimuthal components, i.e., $u_r$ and $u_\theta$ (Fig 3A). Fig 3B shows the spatial distribution of the absolute local velocity $\|u\|$, normalized by the inlet velocity $u_{in}$, while that of the normalized azimuthal flow velocity, $u_\theta/u_{in}$, is depicted in Fig 3C.

To evaluate the perfusion efficiency of each structure, the entire vascular network was divided into two regions, i.e., angiogenic front and inner plexus (Fig 3D and 3E). The angiogenic front was defined as the region of $0.7R_{max} < r < R_{max}$, where $R_{max}$ is the distance of the farthest blood vessel from the origin. The current definition of the angiogenic front is based on the previous observation that c-Myc expression and the proliferation of ECs are active in the region [17,18]. The azimuthal flow rates $Q_\theta$ for the two regions were separately calculated through volume integration of $u_\theta$. This way, the portion of the blood flow transported to the angiogenic front can be quantitatively evaluated. While the vessel density was increased in the *Foxo1* iΔEC mice, $Q_\theta$ at the angiogenic front was significantly decreased in *Foxo1* iΔEC mice (Fig 3F and 3H). In contrast, it was increased for *Prkci* iΔEC mice compared to the control littermates, although vascular branching was decreased (Fig 3G and 3I).

According to Fig 3F and 3G, it can be seen that the blood velocity in the vicinity of the angiogenic front can be four-order-of-magnitude smaller than the inlet velocity (Note that the color contours are logarithmic). However, it does not mean there is no effect of the blood flow at the angiogenic front. Considering the molecular diffusion coefficient of the oxygen, the typical time-scale needed for oxygen to be diffused within a distance of 100 micrometer is in the order of ten seconds. Therefore, if there exists blood flow within a certain distance, oxygen can be supplied to the tissue through the combination of blood flow and molecular diffusion. We can also estimate the significance of the convection by the Peclet number, which is the ratio of the convection and diffusion effects. The Peclet number is around unity even when the blood velocity is 1% of the inlet velocity. This means that the convective effects have similar impacts to the molecular diffusion even when the blood velocity is around 1% of the inlet velocity.

Since the current simulation assumes the blood to be incompressible, the azimuthal flow rate $Q_\theta$ from the artery to the vein has to be exactly compensated by the decrease of the radial velocity so as to satisfy the mass conservation described by Eq (3). We also confirmed that the distribution of $u_r$ is quite similar to that of $\|u\|$ shown in Fig 3B. Specifically, $u_r$ is decreased slowly from the inlet artery toward the angiogenic front when $Q_\theta$ is large, while it decays rapidly when $Q_\theta$ is small.

We also note that there is a substantial difference in $Q_\theta$ between the two control populations shown in Fig 3H and 3I. Using two different mouse lines to obtain the hyper- and hypo-branching phenotypes causes variations in litter and animal sizes. During retinal angiogenesis from postnatal day 0 (P0) to P7, the blood vessel expands rapidly and extensively on the surface of retina. In other words, the vasculature dramatically grows even in a couple of hours, which could cause huge variations in vascular morphology across different mouse lines even in wild type animals. Due to the above reasons, meaningful conclusions can be drawn only through comparisons between KO and control animals.

## 4. Experimental validation

The distinct blood flow distributions for *Foxo1* iΔEC and *Prkci* iΔEC mice should have significant impacts on their transport properties. To validate our numerical results, we observed VEGF expression with an anti-VEGF antibody in the mutant retinae, which reflects the hypoxic status of the tissue. During angiogenesis, VEGF is expressed around the angiogenic front. Once the tissue is vascularized, local hypoxia is improved by oxygen supply with the newly formed blood vessels to downregulate VEGF expression [1].

Since the experiments with each of the hyper- and hypo-branching mutant groups were carried out independently, the basal signal intensity detected for VEGF-A was different across the mutant groups. Therefore, the mean gray value of the VEGF-A signal at the angiogenic front was normalized by the average mean gray values read from the regions showing background noise, e.g. the torn parts of the tissue or the ONH hole in the middle.

Consistent with the decreased flow rate at the angiogenic front in the *Foxo1* [iΔEC] mice, VEGF expression around the angiogenic front is significantly increased compared to that of the control (Fig 4A and 4C). Conversely, VEGF expression in *Prkci* [iΔEC] mice around the angiogenic front region was decreased compared to that of the control (Fig 4B and 4D).

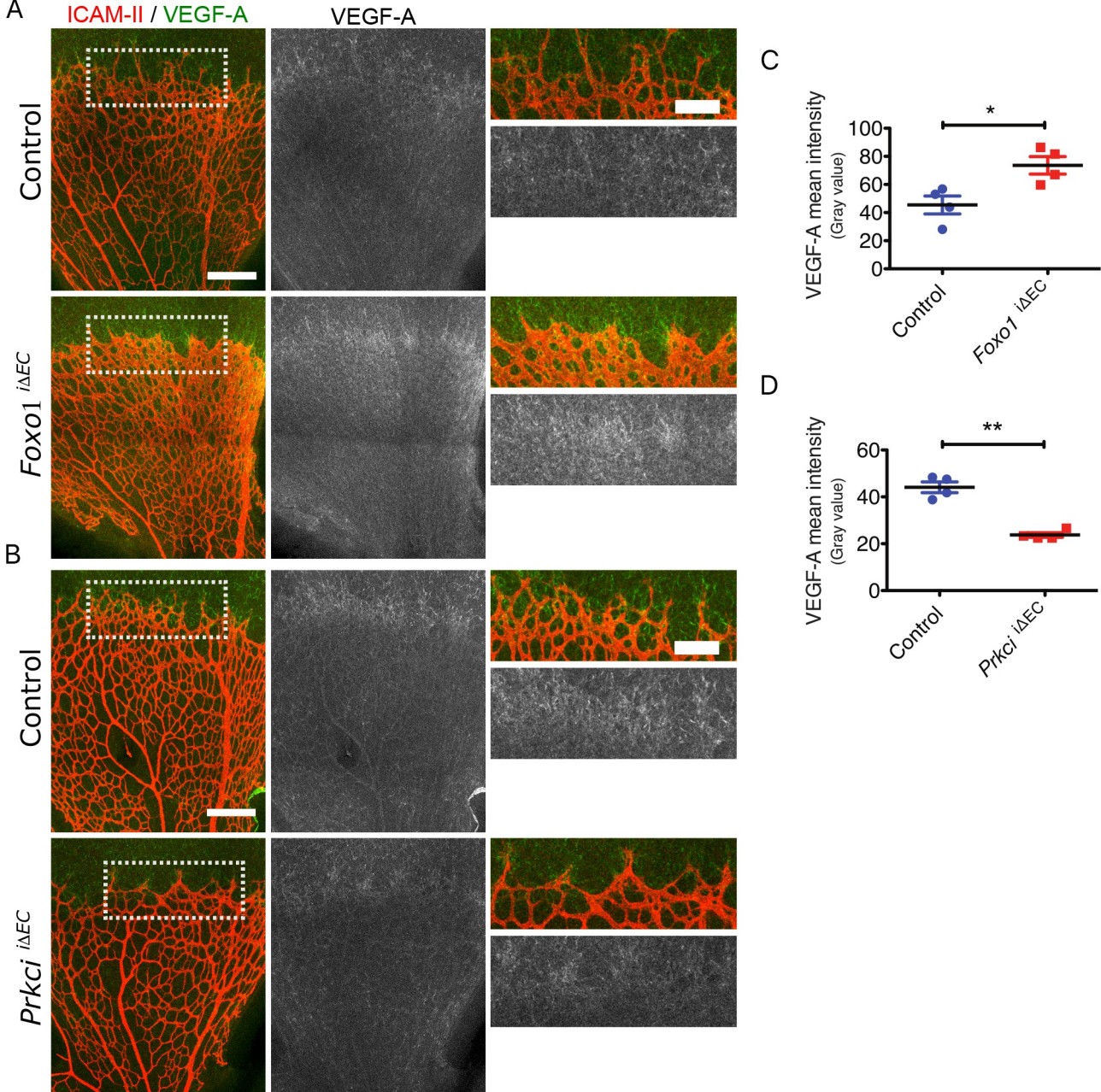

**Fig 4. The effect of the hyper- or hypo-branching vasculature on VEGF expression during angiogenesis A)** Staining of ICAM-II (red) and VEGF-A (green and gray) in the *Foxo1* [iΔEC] and **B)** *Prkci* [iΔEC] mouse retina at P5 and the respective controls; Scale bars represent 200 μm. Higher magnification images of the indicated areas in the left panel are shown in the right panel; Scale bars represent 100 μm. **C)** Quantification of the normalized VEGF-A signal intensity at angiogenic front in *Foxo1* [iΔEC] and **D)** *Prkci* [iΔEC] mutant mouse retinas and the respective controls at P5. Data represent mean ± S.E.M. two-tailed unpaired t-test $^*$p < 0.05 $^{**}$p < 0.01 (n = 4).

## 5. Discussion

While suppression of blood vessel formation has been expected to have a negative impact on blood supply, our results indicate the opposite trend. Namely, the suppression of vessel branching enhances blood flow, and thereby oxygen transport at the angiogenic front. Conversely, the enhancement of blood vessel formation attenuates blood flow at the angiogenic front. The present results underline the importance of detailed flow analysis considering a complex microvascular structure for evaluating its transport properties. In clinical applications, it has been reported that combining anti-angiogenic drug and chemotherapy statistically improves the progression-free survival rate [36]. This suggests that the normalization of the vascular network around tumors could contribute to delivering the medicine to the tumors by enhancing blood supply. Recently, the nanoparticulate drug delivery system has attracted much attention due to its potential to further improve the therapeutic efficacy. Since it is known that the transport of nanoparticles in vasculature is significantly affected by their sizes and shapes [37], the detailed analysis of blood flow and associated mass transfer becomes ever more needed. Computational fluid dynamics of blood flow in vascular networks should be a key for optimizing the shape, size and chemical properties of nanoparticles in drug delivery and even for the design and control of microrobots for future medical applications [38].

Although flows in large vessels such as a coronary artery can be essentially modelled as homogeneous Newtonian fluid, flows in the microcirculation are strongly affected by the complex interactions among plasma flow, interstitial flow, complex geometry of branching patterns and the dynamics of blood cells whose size is close to the blood vessel diameter [39]. As discussed in Sec. 2.5., the Newtonian assumption is considered reasonable, at least, within a single vessel according to existing literatures. However, these evidences have been obtained in relatively straight vessels sufficiently away from branching points [40]. In contrast, near a branching point, the effects of three-dimensionality and axial flow development could be important. However, these effects are neglected in 1D analyses [41,42]. It is also known that red blood cells are not evenly distributed from a parent vessel to daughter vessels. This so-called skimming effect [43] reduces the hematocrit value of a smaller vessel from that of large artery and vein. Therefore, even though the Newtonian assumption is reasonably well for each branch, the effective viscosity could be gradually changed depending on the generation of a branch from the artery or vein.

In order to accurately reproduce such multi-scale and multi-physics phenomena, 3D flow simulation is necessary. Due to its large computational cost, however, most existing studies rely on simplified 1D analysis [41,44], whereas studies applying 3D analysis for vascular network are still limited [45]. In the present study, a new approach to implement a complex structure of vascular network into 3D flow simulation was introduced. By representing an arbitrary complex 3D network structure with the level-set function, the structure was immersed in 3D cartesian coordinate by the volume penalization technique. This has two major advantages: First, the grid generation which is required in flow simulation with a body-fitted coordinate system can be omitted, so that it becomes quite straightforward to simulate flows in different geometries. Second, since the present scheme deploys structured grids in both vessel and tissue regions, the coupling between the blood and interstitial flows and mass transport between blood and tissue can be solved in a unified manner. The latter is particularly important when the transport of oxygen or nanoparticles from blood flow to surrounding tissues will be considered in future work.

Although we assume that the gene modification of *Prkci* or *Foxo1* is the primal cause for the morphologic changes in the vasculature, we cannot exclude a possibility that ischemia induced by the inner shunts discussed in S3 Text causes hyper-branching at the angiogenic front.

During vessel growth, it is widely known that ECs are actively proliferating just behind the angiogenic front and around the vein, but not around the artery [46]. Then, ECs migrate toward upstream of the flow and integrated into the artery [47]. c-Myc is one of the major power generators of endothelial proliferation [17]. Previous reports indicate that EC proliferation of *Prkci* EC specific inducible KO mice results in reduced EC proliferation via compromised c-Myc expression [18], while *Foxo1* EC specific inducible KO exhibit enhanced c-Myc expression and increased EC proliferation in both angiogenic front and vein [17,18]. To address the effects of inner shunt formation or arterial/venous formation would be very interesting. For instance, *Pard3* EC specific inducible KO mice show reduced vessel branching and increased vessel pruning, although c-Myc expression is not affected [48]. Additionally, EC migration toward blood flow is compromised. Another interesting example is EC specific integrin b1 KO mice, which showed attenuated EC migration [49]. The application of the current approach to those two models would be an interesting avenue in future studies.

## Supporting information

**S1 Fig. Comparing the flow distribution within A-V and V-A-V structures derived from the same vascular bed. A**) Staining of ICAM-II in a wild-type mouse retina at P5. White dashed line indicates the retinal tissue border. Red and blue arrows show the retinal radiating arteries and veins, respectively; Scale bar represents 500 μm. **B**) Staining of ICAM-II in one lobe of a wild-type P5 retina from which the A-V and V-A-V structures were extracted. White and green dashed lines indicate the tissue border and isolated V-A-V region, respectively; Scale bar represents 200 μm. **C**) Black and white A-V and **D**) V-A-V structures derived from the same vascular bed (B) for flow simulation. **E**) Visualization of the velocity amplitude on the central *x-y* plane (along the *z* axis) for the A-V and **F**) the V-A-V structures; Color scale represents the logarithmic form of the normalized velocity ($\log_{10}\|\boldsymbol{u}\|/u_o$), where $u_o$ is the outlet velocity in the right vein.
(TIF)

**S2 Fig. Visualization of the velocity distributions around the angiogenic front obtained with (left) current and (right) finer grid resolutions. A**) *Foxo1* CTRL, **B**) *Foxo1* [iΔEC], **C**) *Prkci* CTRL, **D**) *Prkci* [iΔEC]; Color scale represents the logarithmic form of the normalized velocity ($\log_{10}\|\boldsymbol{u}\|/u_{in}$).
(TIF)

**S3 Fig. Visualization of the velocity distribution in two more sets of mutant mice with hyper-branching phenotype. A**) Visualization of the normalized amplitude on the central *x-y* plane (along the *z* axis) for the second and **B**) the third sets of control and *Foxo1*[iΔEC] retinas each from different litters; Color scale represents the logarithmic form of the normalized velocity ($\log_{10}\|\boldsymbol{u}\|/u_{in}$).
(TIF)

**S4 Fig. Visualization of the velocity distribution in two more sets of mutant mice with hypo-branching phenotype. A**) Visualization of the velocity amplitude on the central *x-y* plane (along the *z* axis) for the second and **B**) the third sets of Control and *Prkci*[iΔEC] retinas each from different litters; Color scale represents the logarithmic form of the normalized velocity ($\log_{10}\|\boldsymbol{u}\|/u_{in}$).
(TIF)

**S5 Fig. Effects of morphological changes on the flow rate at different regions. A**) Decomposition of the entire vascular network into three regions, i.e., I: inner region (0–30%), II: middle

region (30–70%), III: outer region (70–100%). **B**) The averaged azimuthal flow rates at the three regions for the *Foxo1* and **C**) *Prkci* mutants and the respective controls at P5. In (B) and (C), the red and blue bars represent the results of mutant and control cases, respectively.
(TIF)

**S1 Table. The ratio of the averaged flow rates in the inner plexus and angiogenic front regions before and after grid refinement.**
(DOCX)

**S1 Data. The simulation codes.**
(GZ)

**S1 Text. Validity of considering a minimum flow unit.**
(DOCX)

**S2 Text. Grid convergence study.**
(DOCX)

**S3 Text. Effects of morphological changes to flow rates at different regions.**
(DOCX)

## Acknowledgments

The authors gratefully acknowledge former graduate students, Mr. Fumiki Mochida, Mr. Chang Cui and Mr. Mingqian Ding, at Institute of Industrial Science, The University of Tokyo for their contributions to developing the software used in this study.

## Author Contributions

**Conceptualization:** Yosuke Hasegawa, Masanori Nakayama.

**Data curation:** Fatemeh Mirzapour-Shafiyi, Yukinori Kametani, Takao Hikita.

**Formal analysis:** Fatemeh Mirzapour-Shafiyi, Yukinori Kametani.

**Funding acquisition:** Yosuke Hasegawa, Masanori Nakayama.

**Investigation:** Fatemeh Mirzapour-Shafiyi, Yukinori Kametani.

**Project administration:** Takao Hikita, Masanori Nakayama.

**Supervision:** Yosuke Hasegawa.

**Writing – original draft:** Yosuke Hasegawa, Masanori Nakayama.

**Writing – review & editing:** Yosuke Hasegawa, Masanori Nakayama.

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
