## [Decision Letter · Decision Letter 0]

17 Nov 2020

Dear Dr nakayama,

Thank you very much for submitting your manuscript "Numerical evaluation reveals the effect of branching morphology on vessel transport properties during angiogenesis" for consideration at PLOS Computational Biology.

As with all papers reviewed by the journal, your manuscript was reviewed by members of the editorial board and by several independent reviewers. In light of the reviews (below this email), we would like to invite the resubmission of a significantly-revised version that takes into account the reviewers' comments.

We cannot make any decision about publication until we have seen the revised manuscript and your response to the reviewers' comments. Your revised manuscript is also likely to be sent to reviewers for further evaluation.

Sincerely,

Daniel A Beard

Deputy Editor

PLOS Computational Biology

Daniel Beard

Deputy Editor

PLOS Computational Biology

Reviewer's Responses to Questions

**Comments to the Authors:**

Reviewer #1: This manuscript supplies a simulation of blood flow in a murine retinal vascular network obtained from microscopy images, to determine how vascular branching affects blood flow in angiogenesis. Interestingly, results indicate that hyperbranching in the network leads to reduced flow, and hypobranching leads to increased flow, during angiogenesis. This simulation result is implicitly supported by experimental results indicating increased VEGF expression due to lower oxygenation in the angiogenic front of the hyperbranching vasculature.

Minor issues:

* It would be nice to see a paragraph in the Introduction and possibly also in the Discussion about other studies simulating blood flow (and possibly oxygenation) in the microcirculation -- what exactly has been done before? On line 73-74, the manuscript says "the effects of altered vascular branching on blood flow distribution and the oxygen transport property have not been investigated yet", but there are plenty of modeling studies that attempt to predict blood flow in a branching network. It would be nice to have some more details about how these are being improved upon in this study, at least mentioning that this study may be the first to use this particular realistic mouse retinal network during angiogenesis.

* In the first paragraph of the "Numerical simulation of blood flow" section (line 115), it is explained why a Newtonian assumption is made about the blood flow; however, in the Discussion, starting with line 241, there appears to be an argument for why flows in the microcirculation cannot be represented in this way. It would be good to have a reminder in the Discussion as to why it's acceptable to assume constant viscosity in this case, and/or describe the limitations of this assumption on the model predictions.

* The ordering of section titles is a bit difficult to follow; it seems to go Introduction, Results, Discussion, Materials and Methods -- although part of the Results section seems to include some Methods. I would suggest moving Materials and Methods to after the Introduction, and possibly even include all of the following subsections: "Image Acquisition", "3D reconstruction", and "Numerical simulation", since these all appear to describe Methods. Then have "Simulation results" and "Experimental validation" as subsections of a Results section.

Very minor things:

* Page 6, line 134: "define" should be "defined"

* Page 7, line 136: I think there should be a comma after the word "time-scale"

* Page 7, line 138: "pulsations" should be "pulsation's"

* Page 9, line 202: The line "Here, Rmax is the distance of the farthest blood vessel from the origin" can be removed, since it appears just above, in lines 199-200.

Reviewer #2: Review of "Numerical evaluation reveals the effect of branching morphology on vessel transport properties during angiogenesis" by Mirzapour-shafiyi et al.

In summary - I think this is an interesting study, but I don't think the interpretation/claims are yet supported by the evidence provided. I would be interested to see a revised submission addressing these critiques.

1. interpretation of azimuthal and radial blood flow

Because all tissue vascular beds are different, it took me a little while to work out why u_theta is being used as the relevant velocity for delivery of blood flow to the periphery rather than u_r, and if I am correct I think the paper would benefit from further description. I think it is as follows: the simulated tissue wedge includes both an artery and a vein (the two large radial vessels on either side of the wedge); since flow should go from artery to vein, u_theta as the flow "around the circle" measures the 'direct' route from artery to vein via peripheral tissue. I don't think that's explained in the manuscript. Fig 2D and 3A note the artery/vein pair, but the choice of flow unit and interpretation is not really described, as far as I can tell.

But also, let's unpack that: the retinal structure is alternating A-V-A-V around the circle. The flow in an artery does not go only to one vein; there is another vein on the other side to which (presumably) half the flow would go. I don't think it will substantially change the findings here, given the normalization being done for the analysis, but the A-V wedge as chosen is not really a full flow unit.

Given the difference between ||u|| and u_theta, we are left with a question as to what u_r represents. Might it be a metric of heterogeneity of flow delivery to the front (i.e. a metric of how much local branches have to compensate for unbalanced distribution - though one would have to take care with sign of velocity when integrating), or maybe a more 'bulk' interpretation where if azimuthal blood flow is low, radial flow would be higher to try to supply that region from inner regions? It might depend on whether we're looking at u_r in the peripheral, inner, or boundary regions. Note also that u_r is not graphed - perhaps it is too similar to ||u||? If so, this should be stated.

Also, can you explain why the normalized azimuthal flow rate is substantially different between the two control populations (Fig 3G vs 3H)?

2. Modeling approach

The modeling engine itself is reasonably well explained; but some questions remain about the choice of modeling approach, including: does the 3d analysis improve on 1d flow analysis? A 1d analysis should be able to provide u_r and u_theta values (and Q values) based on the orientation and diameters of the vessel segments, so the comparison should be made given that this manuscript is introducing an alternate approach and claiming that it is superior.

Also, describe the calculation of Q. Looking at the velocity graphs, there is a four-order-of-magnitude variation in velocity graphed, and it looks like many values at the lower end are really stagnant flow tubes. Do these unbalance the calculation? They should not play a role in overall blood flow.

It would also be helpful to be able to see the code shared for the various steps here, especially since a different method is being used than is typical.

3. Interpretation of causality

If the azimuthal flow is reduced, then the blood needs an alternate path to get from artery to vein. It looks like (Fig 2E, F) this is probably happening via large (or larger) diameter shunts in the inner network. To what extent is the lower/higher flow in the periphery a result of different network architecture in the periphery vs. different network architecture in the inner network?

Related to this, the paper claims that hyperbranching in the periphery is leading to the lower flow, leading to VEGF increase (from the abstract: "hyper-branching morphology attenuates effective blood flow at the angiogenic front and promotes tissue hypoxia"); I think it *much more likely* that inner shunts lead to lower peripheral flow, leading to VEGF increase, leading to hyperbranching. It seems much more likely to be this order, and matches with our understanding of other retinal syndromes, e.g. ROP. How to tell the difference between these causal orderings? I can think of at least one possibility: hybrid simulations with the inner network of one mouse type and the peripheral network of a different mouse type (e.g. inner control/periphery Foxo1iDEC, and vice versa). I think you also need to consider the impact of the artery and vein architecture themselves - e.g. narrowing of the artery as it moves peripherally would favor shunting.

4. VEGF intensity validation

While the raw intensity numbers in the region of interest at the angiogenic front seem different among the various cases, I think some normalization may be in order. for example, it looks like intensity overall in the Foxo1iDEC case is higher throughout the retina, and intensity throughout is lower in the PrkciiDEC case. Given that the conclusion is about the angiogenic front, it would seem reasonable to also compare other regions of interest, e.g. a more central region away from the front, to see whether the flow there (which is more similar among the cases) correlates with local VEGF production.

Minor

Caption to Fig 3 notes n>=3; Fig 4 notes n>=4; but table 1 lists all models and it looks like n=3 for each group (same is noted in the methods section). Please confirm and use explicit n values for each group.

Fig 3E, 3F are the same networks as Fig 2E, 2F. It would be interesting to see some of the other retinal networks that were simulated.

Is there a reason only one flow region (wedge) was chosen for each mouse? How was this region chosen, both in terms of its size and location? Multiple regions per mouse would improve confidence in the predictions.

**Have all data underlying the figures and results presented in the manuscript been provided?**

Reviewer #1: None

Reviewer #2: **No: **Code is not provided; digitized networks not provided; images for most retinas/simulated networks not provided

PLOS authors have the option to publish the peer review history of their article (what does this mean?). If published, this will include your full peer review and any attached files.

Reviewer #1: No

Reviewer #2: No
---

## [Decision Letter · Decision Letter 1]

28 Feb 2021

Dear Dr nakayama,

Thank you very much for submitting your manuscript "Numerical evaluation reveals the effect of branching morphology on vessel transport properties during angiogenesis" for consideration at PLOS Computational Biology. As with all papers reviewed by the journal, your manuscript was reviewed by members of the editorial board and by several independent reviewers. In light of the reviews (below this email), we would like to invite the resubmission of a revised version that takes into account the reviewers' comments.

The biggest concern that of availability of the model code. I agree with Reviewer 2 that you have not provided a strong rationale for not doing so. Furthermore, he reviewer has some additional questions and concerns that you should  consider addressing.

We cannot make any decision about publication until we have seen the revised manuscript and your response to the reviewers' comments. Your revised manuscript is also likely to be sent to reviewers for further evaluation.

Sincerely,

Daniel A Beard

Deputy Editor

PLOS Computational Biology

Reviewer's Responses to Questions

**Comments to the Authors:**

Reviewer #1: My concerns have been adequately addressed

Reviewer #2: The authors have done a good job replying to the comments and critiques overall. While there are some additional simulations I'd love to have seen for scientific interest reasons, I don't think that their absence stands in the way of publication.

I will just reiterate one of my comments, and clarify another, both of which I will leave to the judgment of the editor:

1. The simulation code should be shared. While commenting and documentation are best practice, their absence should not stand in the way of sharing the code. This would aid review and facilitate reproducibility. It is absolutely standard in experimental sciences that experimental reagents are shared, and codes and models are essentially our reagents.

2. In the abstract, the following statement still seems not fully supported:

"Interestingly, hyper-branching morphology attenuates effective blood flow at the

angiogenic front and promotes tissue hypoxia."

I accept the authors' argument that in the critique I over-interpreted this as claiming that hyperbranching *in the periphery* was reducing flow - rather than the hyperbranched network overall - and the expanded discussion of the impact of inner & outer network regions helps to clarify this. I still think that this sentence in the abstract claims tissue hypoxia effects that haven't been measured here (though highly likely) - and which of course could result in further branching. Perhaps '...attenuates effective blood flow at the angiogenic front, likely promoting tissue hypoxia'?

**Have all data underlying the figures and results presented in the manuscript been provided?**

Reviewer #1: None

Reviewer #2: **No: **While the authors state that the data will be provided upon acceptance, the code apparently will not.

PLOS authors have the option to publish the peer review history of their article (what does this mean?). If published, this will include your full peer review and any attached files.

Reviewer #1: No

Reviewer #2: No
---

## [Editor Report · Decision Letter 2]

29 Apr 2021

Dear Dr nakayama,

We are pleased to inform you that your manuscript 'Numerical evaluation reveals the effect of branching morphology on vessel transport properties during angiogenesis' has been provisionally accepted for publication in PLOS Computational Biology.

Best regards,

Daniel A Beard

Deputy Editor

PLOS Computational Biology

Daniel Beard

Deputy Editor

PLOS Computational Biology

---

## [Editor Report · Acceptance letter]

11 Jun 2021

PCOMPBIOL-D-20-01829R2 

Numerical evaluation reveals the effect of branching morphology on vessel transport properties during angiogenesis

Dear Dr Nakayama,

I am pleased to inform you that your manuscript has been formally accepted for publication in PLOS Computational Biology. Your manuscript is now with our production department and you will be notified of the publication date in due course.

With kind regards,

Agota Szep
